# Pulp Mineral Content of Passion Fruit Germplasm Grown in Ecuador and Its Relationship with Fruit Quality Traits

**DOI:** 10.3390/plants11050697

**Published:** 2022-03-04

**Authors:** William Viera, Takashi Shinohara, Iván Samaniego, Naoki Terada, Atsushi Sanada, Lenin Ron, Kaihei Koshio

**Affiliations:** 1Faculty of International Agriculture and Food Studies, Tokyo University of Agriculture, Sakura gaoka 1-1-1, Tokyo 156-8502, Japan or william.viera@iniap.gob.ec (W.V.); nt204361@nodai.ac.jp (N.T.); a3sanada@nodai.ac.jp (A.S.); koshio@nodai.ac.jp (K.K.); 2Santa Catalina Research Site, National Institute of Agricultural Research (INIAP), Panamericana sur km 1, Cutuglahua 171107, Ecuador; ivan.samaniego@iniap.gob.ec; 3Faculty of Veterinary Medicine and Zoothecnics, Universidad Central del Ecuador (UCE), Quito 170521, Ecuador; ljron@uce.edu.ec

**Keywords:** correlation, macronutrients, micronutrients, peel thickness, Passifloraceae, vitamin C

## Abstract

There are several species of passion fruit grown in South America. However, there is a lack of information about the mineral content in their pulp. Thus, the objective of the present research was to determine the mineral content in the pulp of different germplasms of passion fruit [*Passiflora edulis* f. *flavicarpa* (INIAP 2009 and P10), *P. alata* (Sweet passion fruit), *P. edulis* f. *edulis* (Gulupa) and *Passiflora* sp. (Criollo POR1 and Criollo PICH1)] grown in Ecuador and to determine their relationship with relevant fruit quality traits. The results showed that high Mg content was associated with less peel thickness, soluble solids was negatively related to K and B content, and vitamin C was negatively related to S content. INIAP 2009 had high titratable acidity and fruit weight but low N and Na; P10 showed the highest contents of N, K, Na, Mn and fruit weight but less P, Mg, and Fe; sweet passion fruit showed high S, Zn, Cu, soluble solids, and peel thickness but low K, Ca, B, and titratable acidity; Gulupa had high Mg, B, and Zn but low S, Fe, and Mn; Criollo POR1 showed high N and Fe but low Zn; and Criollo PICH1 showed high P, Ca, Mg, and Cu but low soluble solids and peel thickness. These results provide additional information on passion fruit germplasm grown in Ecuador and constitutes a reference for further breeding programs.

## 1. Introduction

There are various species of passion fruit (*Passiflora* spp.) commercially cultivated around the world. Among these species, yellow passion fruit (*P. edulis* f. *flavicarpa* Degener), purple passion fruit (*P. edulis* f. *edulis* Sims), and giant passion fruit (*P. quadrangularis* L.) are the most grown [1], albeit in less proportion the sweet passion fruit (*P. alata* Dryander) [2]. However, there are local germplasms that are underutilized and should be exploited.

*P. edulis* f. *flavicarpa* is called yellow passion fruit (maracuyá or maracuja) [3]. This species is the most cultivated and it is grown in 90% of the orchards in Ecuador, Brazil, and Peru. The fruit measures 6 to 7 cm in diameter and 6 to 12 cm length and its juice is acidic and aromatic [4]. The variety INIAP 2009 (generated from improved genetic material introduced from Brazil to Ecuador) have reached a yield of 20 t ha^−1^, average fruit weight of 174 g, peel thickness of 7.4 mm, and soluble solids content from 13.3 to 14.9. °Brix and titratable acidity varies from 3.4 to 3.9% [5].

*P. edulis* f. *edulis* is called purple passion fruit or Gulupa [3] and it is mainly grown in Colombia. It has a yield of 20 t ha^−1^ [6]. This fruit is almost round, about 5 cm in diameter, its peel is more smooth than yellow passion fruit, and its color is deep purple when fruit is ripe. It has a white mesocarp, intense yellow pulp, and black and oval seeds [7]. This fruit has a weight between 42 and 68 g [8]. In this species, soluble solids vary from 11.65 to 12.65. °Brix and titratable acidity varies from 3.33 to 4.51% [9].

*P. alata* is known as ‘sweet passion fruit’ and is mainly grown in Brazil. This fruit has an oval shape, yellow or orange peel color, weight between 192 and 243 g, height of 9.6 cm, diameter of 7.1 cm, and soluble solids content of 18.5. °Brix and titratable acidity varies from 0.55 to 1.82% [2,10,11]. It reaches a yield of 52 t ha^−1^ [2]. 

In Ecuador, there are some local germplasms (*Passiflora* sp.) called “Criollos”. They have not been already botanically described but they are currently grown by farmers. It has been reported that the Criollo passion fruit reaches a yield of 7 t ha^−1^, and its fruit has a weight of 93 g and peel thickness of 5 mm [5].

Ecuador is one of the main exporters of passion fruit concentrate in South America. There is a cultivated surface of 13,264 ha with an average yield of 6.06 t ha^−1^ [12] and this fruit can be produced all year around with 95% of the total production being the processed by agroindustry and involving around 10,000 small and medium-size farmers [13]. Major production centers of passion are located in the provinces of Los Ríos, Manabí, Guayas, Esmeraldas, Santo Domingo de los Tsáchilas, and Imbabura. These locations have different climatic conditions that are adequate for passion fruit growing [12,13]. 

Minerals are important for human health, and fruits are considered the main sources for minerals in the diet [14]. However, there is very little information about fruit mineral content in *Passiflora* spp., although passion fruit seems to be a good source of minerals as well as carbohydrates, vitamins, and antioxidants [15,16]. Some of the fruits in the *Passiflora* genus are an important source of magnesium (Mg) and zinc (Zn), but they are low in sodium (Na) [17]. It has been reported that 100 g of frozen yellow passion fruit pulp contains 10 mg of Mg, 15 mg of phosphorous (P), 0.3 mg of iron (Fe), 228 mg of potassium (K), and 5 mg of calcium (Ca) [18]. Both yellow and purple passion fruit contain Mg, K, Na, and Zn [17]. Sweet passion fruit contains high P and Fe content, and its consumption contributes the most to the recommended daily intake for these nutrients [19]. The same author also mentioned that the latter species contains K levels comparable to some banana varieties. 

The mineral content in passion fruit pulp adds value to promoting its consumption from a nutritional point of view. A few studies have been carried out concerning this (mainly in commercial cultivars), but these studies have not taken into consideration the local germplasm that is also cultivated by farmers.

The objective of this research was to determine the content of macro and micronutrients in the pulp of *Passiflora* germplasms grown in Ecuador and their relationship with fruit quality traits. Consequently, this information adds value to the passion fruit germplasm as a source of mineral content which benefits human health and which can also be used for breeding programs of this fruit crop.

## 2. Results and Discussion

It has been found that minerals such as nitrogen (N), P, K, and Ca are important for fruit quality in passion fruit [20,21,22,23,24]. In this study, relationships were found between the mineral element content and fruit traits [25] and also among minerals.

### 2.1. Univariate Analysis

#### 2.1.1. Plant Yield and Fruit Quality Traits 

Table 1 shows the results for the fruit traits of the different passion fruit germplasms. P10 (breeding germplasm) showed the highest plant yield followed by the variety INIAP 2009, both genotypes have reported good yield [5] but this trait is influenced by the environment. On the other hand, sweet passion fruit showed low plant yield per plant, which is in agreement with Chavarría-Perez et al. [26] who reported low yields for this species. However, the lowest yield was reached by PICH1 which is a local germplasm that is cultivated by few farmers. On the other hand, a negative correlation was found between plant yield and P which indicated that the pulp of genotypes with more yield and also fruit weight contained less P. However, it has been reported that this element promotes crop yield [27]. Consequently, further research is needed to explain the relationship found in this study. In particular, foliar mineral content should be studied further. 

In terms of plant vigor, no nutrient deficiency was observed in the field. It may be inferred the passion fruit genotypes INIAP 2009, P10, Criolla POR1, and sweet passion fruit had an adequate vegetative growth since their yields were similar to those reported yields in other studies in Ecuador and Brazil [5,26]. Gulupa showed lower plant yield than that reported by Rodríguez-Polanco et al. [28] in Colombia, and PICH1 had the lowest plant yield. However, there are no data to compare to the results of the latter genotype. Therefore, it is recommended that studies about fertilization for these two genotypes should be carried out to find nutrient levels to increase their yields, taking into consideration that this trait is highly influenced by the environment conditions. 

INIAP 2009 showed the highest fruit weight which is close to that reported by Álvarez et al. [29] with 230.33 g higher than that mentioned by [5] (with 174.29 g). P10 also had a higher fruit weight than INIAP 2009, and this is reasonable due to the fact that it comes from a segregation process of the INIAP’s variety. Criollo PICH1 and Gulupa showed lighter fruit weights. In other studies, Rojas et al. [7] and Thokchom and Mandal [30] reported higher Gulupa fruit weight (around 40 g) than in this research but lower weight for the yellow passion fruit (around 115 g). On the other hand, a higher polar diameter was obtained by Sweet passion fruit while INIAP 2009 showed a higher equatorial diameter. 

Peel thickness is a trait valued by passion fruit farmers. The lowest value for this trait was obtained for Criollo PICH1 followed by Gulupa. Although INIAP 2009 and Criollo POR1 had higher peel thickness than the genotypes mentioned above, they are the main passion fruit cultivated in Ecuador mainly since they produce bigger fruit. In this study, the peel thickness value obtained for INIAP 2009 is higher than that reported by Viera et al. [5]. However, these authors mention that this trait can vary due to genotype x environment effect.

Sweet passion fruit showed the highest soluble solids content which is similar to those reported by Rinaldi et al. [10] and Santos et al. [2] with 18.57 and 20.00° Brix, respectively, while Criollo PICH1 obtained the lowest value. It has been reported a value of soluble solids of 14.60° Brix for the Criollo genotype [5], which is a little higher than Criollo POR1 in this study. Gulupa reached a similar soluble solids content to that mentioned by Thokchom and Mandal [30] with 14.00° Brix, and a little higher than reported by Granados et al. [31] and Bermeo [9] with 13.00 and 12.65° Brix, respectively. 

In terms of titratable acidity, INIAP 2009 showed the highest value, which is slightly higher than that reported by Viera et al. [5] (with 3.65%). A value of 2.40% has been reported for Gulupa [30], which is similar to that which was obtained in this study. However, Bermeo [9] and Granados et al. [31] found higher values (3.33 and 3.24%, respectively). 

Sugar/acid ratio is an important trait since it is related to fruit taste. According to the results, sweet passion fruit showed the highest ratio, having the sweetest taste. Gulupa could be considered as a sweet/acidic fruit, but all yellow passion fruit genotypes obtained lower values, having an acidic taste (which is the preference for the agroindustry in Ecuador). In passion fruit, it has been reported that elements such as N and Ca positively influence this parameter whereas P has the opposite effect [20,22,24]. 

All genotypes, except for sweet passion fruit, did not show statistical differences in pulp yield, which means that they have good amount of pulp content. However, this trait is proportional to the fruit weight of each germplasm. Similar values have been reported for this trait in INIAP 2009 and genotype Criollo [5].

#### 2.1.2. Mineral Content 

The six germplasm of passion fruit showed statistical differences for all minerals (Table 2). 

In terms of macronutrients, N is an essential element for fruit set and it is related with fruit quality [32]. Criollo POR1 and P10 had the highest N content while INIAP 2009 obtained the lowest value. Nevertheless, Ramos et al. [15] reported the highest content of N (2400.00 mg 100 g pulp^−1^) in the yellow passion fruit.

It has been reported that passion fruit has a high content of K [33]. This element is essential for human health since it plays a role in the normal functioning of cells and organs, being related to blood pressure regulation, muscle contraction, and nerve transmission [34]. P10 had the highest K content which agrees with Carvajal et al. [17] and Ramos et al. [15] who found values of 1440.00 and 3800.00 mg 100 g pulp^−1^, respectively, in the yellow passion fruit. Sweet passion fruit reached the lowest result, which is higher than that found by Da Silva et al. [35] with 740.60 mg 100 g pulp^−1^. However, this value in terms of fresh weight (250.12 mg 100 pulp^−1^ f.w.) is also lower to than that reported by Souza et al. [19] with 375.42 mg 100 pulp^−1^ f.w. P is an essential element for human nutrition and health. It performs vital functions in skeletal and non-skeletal tissues and is pivotal for energy production and other physiological processes [36]. Usually, P is mainly derived from animal protein sources. Hence, high P content in the fruit pulp is highly valued. It has been found that purple passion fruit had less P than the others species [15]. However in this study, the yellow passion fruit (P10) showed the lowest P content while Criollo PICH1 showed the highest content. 

Ca is considered a critical nutrient in determining fruit quality [37]. Criollo PICH1 had the highest Ca content whereas Sweet passion fruit obtained the lowest value. Da Silva et al. [35] found higher values of Ca (56.80 mg 100 g pulp^−1^) in sweet passion fruit. Moreover, this result in terms of fresh weight (1.36 mg 100 g pulp^−1^ f.w.) is also inferior to that one reported by Souza et al. [19] with 4.76 mg 100 g pulp^−1^ f.w. 

Passion fruit is an important source of Mg [17] and it is an essential nutrient for the physiologic functions of various body organs [38]. Gulupa had the highest Mg content which was higher than that found by Ramos et al. [15] and Carvajal et al. [17] with 120.00 and with 90.00 mg 100 g pulp^−1^, respectively, while P10 had the lowest value. Therefore, the consumption of purple passion fruit should be encouraged due to its Mg content, especially in the South American countries where yellow passion fruit is preferred.

Sweet passion fruit had the highest sulfur (S) content, a value larger than that found by Da Silva et al. [35] with 98.3 mg 100 g pulp^−1^, whereas Gulupa obtained the lowest value which was lower than that mentioned by Ramos et al. [15] with 90.00 mg 100 g pulp^−1^. 

It has been mentioned that passion fruit is a low source of Na [17,33] which agrees with the results of this study. P10 had the highest Na content while INIAP 2009 had the lowest value (8.81 mg 100 g pulp^−1^). Both values are higher than that found by Ramos et al. [15] with 1.40 mg 100 g pulp^−1^.

In terms of micronutrients, Gulupa had the highest boron (B) content which was higher than that reported by Ramos et al. [15] with 0.20 mg 100 g pulp^−1^, while sweet passion fruit had the lowest value. The lack of arils (the pulpy tissue surrounding the seed) is a characteristic associated with B deficiency [39]. However, none of the passion fruit germplasm showed this disorder, thus this element would be in adequate amount in the fruit.

Zn is essential element for human health since it is related to the immune system [40], and passion fruit has been reported as good source of Zn [17]. Sweet passion fruit and Gulupa had the highest Zn content. Nevertheless, Ramos et al. [15] found the lowest Zn contents in the purple passion fruit which is opposite to this study. Sweet passion fruit is an underutilized species that should be more consumed due to its medicinal properties [41], even moreso since it showed high Zn content. Gulupa also was high this element and it had high vitamin C content [16], that might be a reason of its preference in Oceania and Asia countries, in addition to its sensory properties [9]. On the other hand, Criollo POR1 had the lowest value albeit values greater than that reported by MEXT [42]. 

Criollo PICH1 and Sweet passion fruit had the highest copper (Cu) content. However higher values (0.30 mg 100 g pulp^−1^) have been reported by Da Silva et al. [35] in the latter species. On the other hand, similar values (between 0.12 and 0.14 mg 100 g pulp^−1^) were found in the rest of the passion fruit germplasm. Cu is an element that plays an important role in the immune system maintenance [43]. According to the results, sweet passion fruit showed high Cu and also Zn content, making this species a good source of these two elements that are related to human immunity [43]. 

Fe is an essential micronutrient for oxygen transport, metabolism, and many enzymatic reactions [44]. Criollo POR1 had the highest Fe content whereas P10 had the lowest value. The latter was slightly lower than that found by Ramos et al. [15] who reported that yellow passion fruit had 5.50 mg 100 g pulp^−1^. The same author [15] also reported low Fe content (2.90 mg 100 g pulp^−1^) in purple passion fruit (lower than that found in this study). 

Manganese (Mn) is present in low concentrations in dietary sources and it is important for physiological processes in the human body [45]. P10 had the highest Mn content while Gulupa had the lowest; the latter was lower than reported by Ramos et al. [15] with 0.40 mg 100 g pulp^−1^.

To sum up in terms of species, *P. edulis* f. *flavicarpa* showed high content of K, Na, and Mn; the results of K and Mn are similar to that reported by Ramos et al. [15]. *P. edulis* f. *edulis* had the highest Mg, B and Zn content. However, these results contradict those found by Ramos et al. [15]. *P. alata* showed high S, Zn and Cu content. This underutilized species should be more exploited for its medical properties [41], characteristics that could be related to its high Zn and Cu content in the fruit pulp. Zn and Cu play significant roles in the immune system. Their consumption might even be a preventive and promising option to enhance human immunity against COVID-19 and its new strains [43]. The local germplasm (*Passiflora* sp.) showed high content of N, Fe (POR1), P, Ca, Cu (PICH1). Thus, this type of native germplasm should be more exploited due to their mineral properties that contribute to human health [43,44].

The results obtained about the mineral content in this study complement the phytochemical characterization and antioxidant activity of the passion fruit germplasms grown in Ecuador [16]. INIAP 2009 showed relatively high K, Ca and Zn, and has high polyphenol content and antioxidant activity. P10 showed the highest contents of N, K, Na and Mn, and also has high polyphenol content and antioxidant activity. Sweet passion fruit showed high S, Zn and Cu, and has high polyphenol content. Gulupa showed high Mg, B and Zn, and has high vitamin C and relatively high flavonoid and carotenoid content. Criollo POR1 showed high N and Fe, and has relatively high carotenoid content. Finally, Criollo PICH1 showed high P, Ca, Mg, and Cu, and has high flavonoid and carotenoid content. Both local germplasms, namely POR1 and PICH1, have low antioxidant activity [16].

### 2.2. Regression Analysis

Pearson coefficients showed relationships both between the mineral element content and fruit traits and among macro and micronutrients (Table 3). Plant yield showed a positive association with K, B, and Mn. These three elements have been related with crop yield since they have roles to play in plant metabolism and the photosynthetic process [46,47,48] and deficiency of them usually reduces productivity. The amount of K required by the change according to the phenological stages of the crop [46] while Mn is required in small quantities [49]. An inadequate boron supply exhibits a detrimental effect on the yield of agricultural plants [47] but it is solved by foliar fertilization.

Fruits with greater weight showed less Mg, results opposite to those ones obtained in grapevine [50] and pomegranate [51] in relation to this mineral. Nevertheless, it was found that this element did not influence fruit weight in figs [52], which could indicate that this relationship differs according to the fruit species. In the case of passion fruit, this relationship may be explained since Mg decrease ring thickness [53], and this would have a direct influence on fruit weight. 

High pulp yield was positively related to B content. It has been reported that this element increases this fruit trait in guava [54], and this could be associated with the fact that B promotes cell division [55]. This trait was also correlated with K; it has been reported that this element can increase fruit pulp in pineapple, and this could be related to the fact that this element is involved in photosynthetic and metabolic processes that influence fruit quality [56]. 

Vitamin C was negatively related to soluble solids, which means that sweeter fruits will have less of this vitamin. Ascorbic acid influence fruit acidity [57], and acidic fruits have less Brix content. Vitamin C was also negatively related with S content. This effect has been also reported in short cycle crops [58,59], but this is contradictory with the fact that this element is essential for the production of vitamins in plants [60]. S was positively related to soluble solids which is in agreement with the findings of Mostafa [61] in grapevines.

Soluble solids content was opposite to K content, which agrees with that mentioned by Obreza et al. [53] in citrus. Consequently, this element had the opposite relationship with titratable acidity, it being reported that the titratable acidity should be proportional to the K concentration [62]. On the other hand, Carvalho et al. [63] found that K fertilizations increase soluble solids content in pineapple, which could be due to the act that K is important for plant photosynthesis and that this process favors sugars production. This element also regulates sucrose loading [64]. Soluble solids were also negatively correlated with Ca content which is in accordance with that found by Moor et al. [65] in apples. It may be related to Ca as it is involved mor in structural cellular processes [66] than photosynthesis, which can influence directly the sugar production in plants [64]. However, it has been reported that Ca induces the accumulation of numerous soluble sugars such as glucose, fructose, and sorbitol [67]. On the other hand, increasing of N supply can decrease glucose, sucrose, fructose, and total nonstructural carbohydrates in apple [68]. High K could promote photosynthesis and modify the distribution of the carbohydrate from leaves to fruits while low K inhibits carbohydrate metabolism during maturation [69]. 

P negatively influenced titratable acidity, a result which agrees with that which has been reported by Obreza [53] in citrus. In addition, Medeiros et al. [70] and Ahmad et al. [71] also found this effect in strawberry, observing that low concentrations of P produce high titratable acidity in the fruit. 

Peel thickness showed a negative correlation with Mg. This element decreased rind thickness [53], therefore affecting peel (rind + exocarp) thickness.

Synergism and antagonism relationships occur between minerals and they are called inhibition or potentiation relationships. Synergism generally occurs between elements with different valences, while antagonism occurs between those with similar valences [72]. In addition, interactions between nutrients happen when the supply of one nutrient affects the uptake, distribution or function of another nutrient [73]. In terms of significant relationships among minerals, K had positive correlation with Mn and negative with Zn/On the other hand, B and Cu were inversely related. The first association is in accordance with that mentioned by Fan et al. [74] who reported this relationship among these elements in plants, but for these authors K and Zn had a positive association (whereas B and Cu did not show any relationship). In addition, Na and Mn showed a positive correlation, which would mean that there was a synergism effect between these two elements. 

Overall, further research is needed to explain the correlations between physical traits and among minerals found in this study since they may be affected by environmental conditions and passion fruit genotypes.

### 2.3. Principal Component Analysis

According to the principal component analysis (PCA) (Figure 1), the two first components explained 65% of the variance observed in the data. PCA clearly indicated that the first component was a contrast between K, B, and vitamin C vs. S and SS, showing a difference between more acidic fruits vs sweeter fruits; this component was also slightly influenced by N and Fe. The second component was a contrast between Mg vs. peel thickness and slightly influenced by Ca.

Peel thickness is a characteristic for breeding purposes since Ecuadorian farmers want fruit with less peel thickness, but this trait had genotype x environment effect [7] which complicates this purpose. Fruits showing this characteristic showed high Mg content in the pulp, but there was not a clear linear tendency (R^2^ = 0.56), it could be explained since peel thickness vary according to the genotype and the environment conditions where the passion fruit plants are grown. Nevertheless, Mg content may be considered as an indicator trait to be evaluated in further breeding progeny to corroborate this relationship, but also considering that a low heritability of this element has been reported in other fruit crops [75]. 

Mg has been related to fruit quality [50,51] and it has been reported that this element decreases rind thickness [53] which is a component of peel thickness. However, Ca and B are also necessary for peel formation [76]. Studies in other perennial crops have found that fertilization using high dose of Mg can slightly decrease peel thickness [77,78]. However, the Mg fertilization dose will depend on the crop nutrition and soil conditions [79]. Therefore, more research is needed to corroborate the above mentioned and to define the amount of this element to be applied, even depending on the phenological stages of the crop. In this study, all of the soils where passion fruit was grown had high content of Mg (Table 4) but the peel thickness response was also depending on the cultivated species. 

Soluble solid content is an important factor since the price of the passion fruit in the agroindustry increases as fruit Brix degrees increase. For this reason, this parameter is very important for farmers. It has been found that Ca had an influence on increasing soluble solid content [24]. However, the results obtained in the PCA analysis in this study were the opposite; the same authors mention that the effect of this element can vary depending on the type of fruit. Moreover, soluble solids negatively influenced vitamin C content, one of the most appreciated characteristics of the tropical fruits [81]. Gulupa had high vitamin C content [16] which was relatively higher than that reported by Granandos et al. [31] with 25.50 mg 100 g pulp^−1^; the latter value was similar to that one observed by INIAP 2009 [16]. Lower values for vitamin C have been reported in INIAP 2009 and genotype Criollo [5].

K is highly related to fruit quality traits such as soluble solid content and titratable acidity [82] as observed in this study. However, K content should not be a considered as a target for breeding purposes of passion fruit due to their negative influence on soluble solids found in this study, which is in agreement with the results mentioned by Bashira et al. [51]. Moreover, it also showed negative correlation with Zn which is an important element in terms of health properties. In addition, it has been reported that K fertilization increase this parameter in strawberry fruit [83]. On the other hand, it has been reported that B influence negatively to SS content [84] as found in this study. This could be due to B transports sugar through the formation of borate-sugar complexes in higher plants [85].

It has been reported that low K content reduce titratable acidity [21] while Ca is negatively associated with this trait [24], which agrees with the results of this research (Figure 1). In addition, an excess of P can cause high fruit acidity which deteriorates passion fruit quality [22]. 

Fruits with less Zn and Cu showed higher titratable acidity; these two minerals have been related to minimum titratable acidity [86,87,88].

Gulupa and Criollo PICH1 were associated with Mg content and consequently with less peel thickness. This result is interesting since these two genotypes are phenotypically distinct (Gulupa is purple passion fruit while PICH1 is yellow passion) and grown in distinct altitudes which means different climatic and soil conditions, but in both cases the Mg content in the fruit was high. INIAP 2009, P10, and Criollo POR1 (all yellow passion fruit) showed different sizes of peel thickness even though they were grown in the same site, which means that this trait varies according to the genotype and environment conditions [5]. On the other hand, INIAP 2009 was associated with titratable acidity, P10 with Mn content, sweet passion fruit with soluble solids and S content, and POR1 with N content. 

In terms of breeding, P10 and POR1 may be considered as parental for further hybridization since the former has large fruit size while the latter had less peel thickness, the traits most appreciated by producers. In addition, both had good content of some antioxidant compounds [16], minerals such as K, and soluble solids content that are important for fruit quality. Also, both are yellow passion fruit (preference of the Ecuadorian farmers). 

This research has generated information about the mineral content of commercial passion fruit (yellow and purple), but has also considered underutilized (*P. alata*) and local germplasms which, despite having a smaller fruit size and less yield, have been shown to be a source of minerals in the fruit pulp (particularly S, Cu, Zn, and P). Sweet passion fruit showed high Zn and Cu content, a characteristic that would be related to its medicinal properties. 

Passion fruit species have different environmental requirements for their adequate vegetative growth and production [89]. Although a limitation of this research is that the passion fruit germplasms were grown in different Ecuadorian environmental conditions (i.e., altitude, precipitation, and heliophany) due to their preference in adaptation for their growth and production, the results of this study are useful as a reference for further fruit chemical characterizations of passion fruit species, as well as for breeding programs and genotype x environment research.

## 3. Materials and Methods

### 3.1. Experimental Site and Plant Material

The experiment was conducted on 2021 and it was carried out at the Nutrition and Quality Laboratory (ISO/IEC 17025) of the National Institute of Agricultural Research (INIAP), located in Cutuglahua, province of Pichincha (00°22′57″ S, 78°33′18″ W). 

Passion fruit species belong to the ex situ germplasm collection of INIAP. Six genotypes belonging to different species were evaluated. In terms of *P. edulis* f. *flavicarpa*, the genotypes were the variety INIAP 2009 which was released in 2009 and is currently cultivated by around 50% of the farmers, and P10 which is a selected genotype (not yet released) from a mass selection process (segregation) carried out in the last two years. The Sweet passion fruit (*P. alata*) and Gulupa (*P. edulis* f. *edulis*) are introduced passion fruit that are grown in much less proportion (around 10%); the former was introduced from Brazil and the latter possibly from Colombia. POR1 and PICH1 are local germplasm (genotypes), the former is the second most commercial passion fruit cultivated after INIAP 2009 while the latter is a genotype just grown by a few farmers. However, all of these genotypes constitute a source of germplasm that could be exploited for their particular characteristics in terms of fruit chemical composition [16]. 

Passion fruit genotypes were propagated by cuttings and were transplanted to the field in August of 2020 and sampling was carried out on February of 2021 in the orchards belonging to the research sites of INIAP (Table 5) where the passion fruit germplasms were grown according to their conditions for adaptation and production. Fruit of the different passion fruit genotypes (Figure 2) were harvested from one-year-old plants at maturity grade 5 (100% color change on the tree) [90,91]. After harvest, fruits were taken to the laboratory where the pulp was extracted, homogenized, and lyophilized. 

Each genotype (germplasm) was set as a treatment (irrespective the place of origin) with the objective of identifying differences in the fruit quality characteristics and mineral content of the accessed germplasms. For the fruit quality variables (n = 25), one fruit from twenty-five different plants was harvested and each fruit was considered as a replication. For the mineral analysis (n = 3), five fruits from five plants were harvested (25 fruits in total) and this was carried out by triplicate using different plants to obtain three independent samples. The experimental unit was constituted by one ripe fruit (grade 5) for the fruit quality traits and 10 g of freeze-dried pulp for the mineral analyses.

Soil mineral content of the sites had similar nutrient content in terms of interpretation (high, medium or low) for most of the elements (Table 4). B was the only nutrient that had low content but this an element that is found in low concentrations in the majority of soils [92]; however, this was overcome with foliar fertilization. Fertilization per plant was carried out as follow: N 450 g, P 45 g, K 240 g, Mg 10 g, S 15 g, Mn 2 g, Cu 0.1 g, Fe 0.5 g, Zn 0.2 g, and B 0.2 g.

### 3.2. Plant Yield and Physical Fruit Traits

The yield per plant was calculated based on the number of fruits and their weight.

Fruit weight was determined with a digital scale (BBC31, Boeco, Hamburg, Germany) and expressed in grams. Fruit diameter and peel thickness (rind and exocarp) were recorded by a digital caliper (CD-8 CB, Mitutuyo, IL, USA) and both were expressed in mm. Peel, pulp, and seed yield was estimated by weighing each part using a digital scale (BBC31, Boeco, Hamburg, Germany) and applying the following formula [93]: $$X=\frac{\mathrm{Wp}}{\mathrm{Wf}\;}\times 100$$
where X is the percentage of each fruit part (peel, pulp or seed), Wp is the weight of each fruit part (peel, pulp or seed), and Pf is the total fruit weight. 

### 3.3. Soluble Solids Determination

Total soluble solids were determined by refractometry using a digital refractometer (N2-E, ATAGO, Tokyo, Japan), according to the methodology specified by Viera et al. [94]. Two drops of passion fruit juice were placed on the prism of the equipment surface and soluble solids were expressed in terms of °Brix.

### 3.4. Titratable Acidity Determination

Titratable acidity was measured by potentiometric titration using a standardized alkaline solution [94]. A total of 30 g of fruit pulp was weighed and mixed with distilled water at a volume of 200 mL. Subsequently, a 20 mL aliquot was placed in a 25 mL beaker and titrated with a 0.1 N NaOH solution until pH 8.2 was reached. The results were reported based on citric acid (%).

### 3.5. Sugar/Acid Ratio (SAR)

This ratio was determined using the relation between the total soluble solids and the titratable acidity [94]. According to the equation the following formula:$$\mathrm{SAR}=\frac{\mathrm{TSS}}{\mathrm{TA}}$$
where TSS is total soluble solids, and TA is titratable acidity. 

### 3.6. Vitamin C 

The values for vitamin C reported by Viera et al. [16] for the six passion fruit genotypes were used for the statistical analysis.

### 3.7. Sampling Preparation for Mineral Analysis

After harvesting, fruits were washed and the pulp, seeds, and peel were separated. The pulp was placed in 250 mL plastic containers, frozen at −12 °C and then subjected to a lyophilization process at −70 °C and 1 bar of pressure. The dry samples were ground in a mill (ZM 200, Retsch, Hann, Germany) until a particle size of less than 1 mL was obtained. Finally, samples were stored in hermetically sealed plastic bottles.

### 3.8. Determination of Total Nitrogen

Total N analysis was carried out by the Kjeldahl method [95], for which 1 g of the sample was weighed in a 250 mL digestion tube, 2 copper catalyst tablets (3.5 g K_2_SO_4_ y 0.4 g CuSO_4_ × 5 H_2_O), 12 mL of concentrated sulfuric acid were added and subjected to a digestion process at 400 °C for 1 h in a digester (DKL 12, Velp Scientifica, New York, USA). The digested samples were cooled and placed in an automatic nitrogen analyzer (Kjeltec 8400, Foss, Hillerod, Denmark) for distillation and titration, for which 60 mL of NaOH 40% and 40 mL of water Type I were added to each tube and the distillation was carried out. The distillate was received in a 3% boric acid solution and titrated with a 0.3 N hydrochloric acid solution. The results were expressed as mg 100 g^−1^ of dry sample.

### 3.9. Sampling Digestion

Samples were subjected to a mineralization process according to the method proposed by AOAC [95], for which 1 g of the dry sample was weighed in a 25 mL porcelain crucible and subjected to an incineration process in a muffle (model 48000, Thermolyne, Dubuque, IA, USA) at 400 °C for 12 h. Subsequently, the crucibles were cooled in a desiccator and transferred to a heating plate (Witeg, Wertheim, Germany); then 5 mL of HCL (37%) and 10 mL of Type I water (18.2 MΩ cm) was added to each crucible, and the samples were digested at 100 °C until the volume was reduced by half. Samples were filtered using qualitative filter paper (Watman, Maidstone, UK), in a 100 mL flask and filled with water Type I.

### 3.10. Determination of Macro and Micronutrients

In the case of P, 0.5 mL of the sample was taken and 4 mL of water Type I and 0.5 mL of ammonium molybdovanadate 1% solution were added. The sample was stirred and the absorbance was measured in a UV-Visible spectrophotometer (UV2600, Shimadzu, Tokyo, Japan). Quantification was based on a calibration curve, and the results were expressed in mg 100 g^−1^ of dry weight sample.

For the analysis of macronutrients, an aliquot of 4.5 mL of sample was taken in a test tube, 0.5 mL of lanthanum 1% solution was added for determination of Ca and Mg. In the case of Na and K, 0.5 mL of lithium 1% solution was added in the same volume of sample and to eliminate interferences. 

For the analysis of micronutrients (Zn, Cu, Fe, and Mn), 5 mL of each sample was taken and no solutions were added for interferences. In the prepared samples, the absorbance was measured in an atomic absorption spectrophotometer (AA7000, Shimadzu, Tokyo, Japan). S and B content were measured directly from the solutions of each sample in an inductively coupled plasma spectrophotometer (5300 Optima DV, Perkin Elmer, Bresia, Italy). The quantification was carried out using calibration curves for each element; the results were expressed as mg 100 g^−1^ of dry weight sample.

### 3.11. Statistical Analysis

For the statistical analysis, each genotype was considered as a treatment. Levenne test was calculated to set the homogeneity of variances. Analysis of variance (univariate analysis) was carried out with all data from fruit quality traits and minerals. A Tukey test at 5% was used to determine differences among means. Data analysis was carried out in the R statistical program version 4.04 [96].

Pearson correlation coefficients (d.f. = 16) were calculated to measure the linear correlation between two independent variables, considering the fruit quality traits and all minerals. 

Principal component analysis (multivariate analysis) was used to visualize the relationship among the relevant fruit quality traits and minerals, and their association with the *Passiflora* germplasm.

## 4. Conclusions

In this study, Mg content was negatively associated with peel thickness. This element could be considered as an indicator trait for further evaluation in breeding populations. Soluble solids content was negatively related to K and B content while vitamin C was negative associated to S content. Sugar/acidity ratio is an important trait for passion fruit taste due to sugars such as sucrose and fructose influence in the perception of fruit sweetness which is reflected in fruit flavor. 

The most relevant traits for each genotype were as follows: INIAP 2009 had high titratable acidity and fruit weight but low N and Na; P10 showed the highest contents of N, K, Na, Mn, and fruit weight but less P, Mg, and Fe; Sweet passion fruit showed high S, Zn, Cu, soluble solids, and peel thickness but low K, Ca, B, and titratable acidity; Gulupa had high Mg, B, and Zn but low S, Fe, and Mn; Criollo POR1 showed high N and Fe but low Zn; and Criollo PICH1 showed high P, Ca, Mg, and Cu but low soluble solids and peel thickness. 

These results add value to the nutrition composition of the passion fruit and can be used as reference for further breeding programs of passion fruit in terms of fruit mineral content and fruit quality traits (and, in particular, to use underutilized or native germplasm to generate new breeding populations). 

On the other hand, it is recommended that researchers carry out further studies about Mg fertilization in passion fruit using high doses of this element to determine its effect in decreasing peel thickness (since this is a desirable trait for farmers). 

## Figures and Tables

**Figure 1 plants-11-00697-f001:**
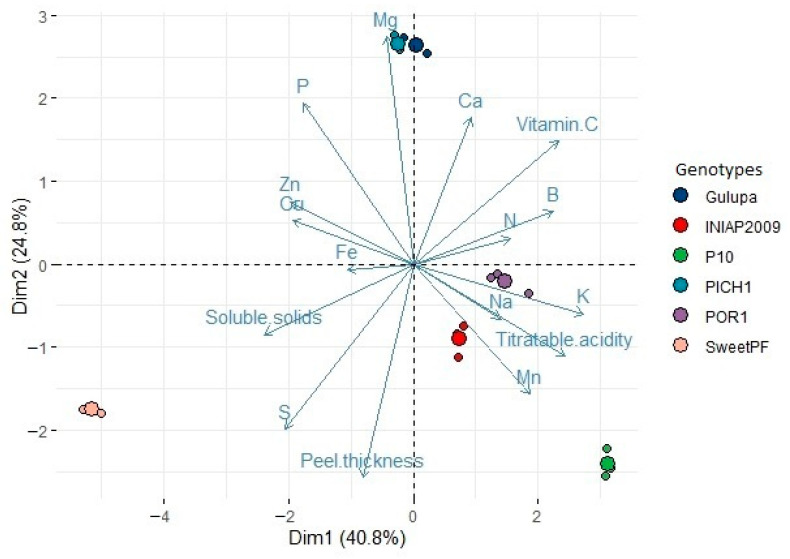
Principal component analysis for relevant fruit quality traits and mineral content of passion fruit germplasm grown in Ecuador.

**Figure 2 plants-11-00697-f002:**
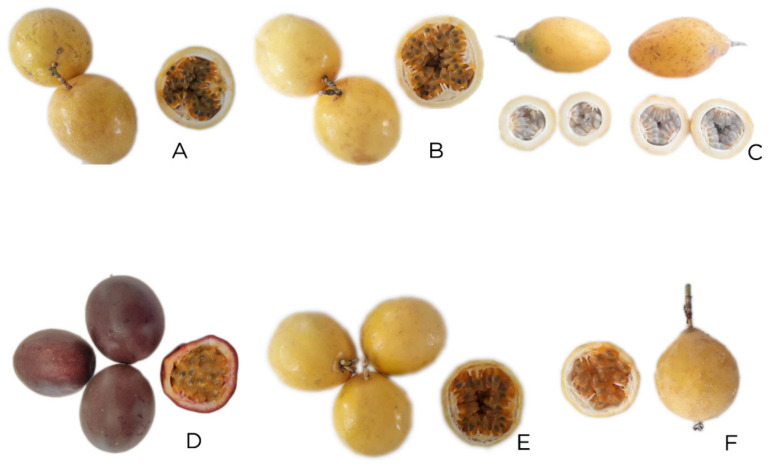
Passion fruit germplasm grown in Ecuador. (**A**) INIAP 2009, (**B**) P10, (**C**) Sweet passion fruit, (**D**) Gulupa, (**E**) POR1, and (**F**) PICH1.

**Table 1 plants-11-00697-t001:** Fruit quality parameters of the passion fruit germplasm grown in Ecuador.

Genotype	Yield (kg Plant^−1^)	Fruit Weight * (g)	Polar Diameter * (mm)	Equatorial Diameter * (mm)
INIAP 2009	15.27 ± 1.42 b	223.63 ± 49.44 a	96.18 ± 7.13 b	90.20 ± 5.53 a
P10	19.45 ± 4.53 a	207.93 ± 63.48 a	96.45 ± 7.69 b	82.05 ± 4.43 b
Sweet passion fruit	3.60 ± 0.67 d	184.66 ± 35.04 b	117.80 ± 6.60 a	67.97 ± 2.19 c
Gulupa	7.64 ± 0.76 c	34.22 ± 3.73 d	49.27 ± 3.41 d	44.13 ± 2.60 d
Criollo POR1	7.15 ± 0.79 c	109.64 ± 15.52 c	76.73 ± 3.76 c	69.72 ± 3.67 c
Criollo PICH1	1.75 ± 0.19 d	32.96 ± 3.76 d	50.03 ± 5.57 d	44.77 ± 3.25 d
**Genotype**	**Peel Thickness * (mm)**	**Soluble Solids Content ** (°Brix)**	**Titratable** **Acidity ** (%)**	**Sugar/acid** **Ratio ****
INIAP 2009	10.38 ± 1.32 b	12.13 ± 0.15 d	4.43 ± 0.02 a	2.71 ± 0.02 f
P10	10.01 ± 1.22 b	12.23 ± 0.15 d	4.24 ± 0.04 b	2.88 ± 0.03 e
Sweet passion fruit	11.54 ± 0.99 a	19.47 ± 0.12 a	1.12 ± 0.01 f	17.41 ± 0.22 a
Gulupa	5.36 ± 0.60 c	14.70 ± 0.10 b	2.07 ± 0.02 e	7.10 ± 0.03 b
Criollo POR1	5.87 ± 0.37 c	13.47 ± 0.06 c	3.91 ± 0.05 c	3.43 ± 0.04 d
Criollo PICH1	3.85 ± 0.44 d	11.00 ± 0.20 e	2.20 ± 0.04 d	5.00 ± 0.15 c
**Genotype**	**Peel Yield *** **(%)**	**Pulp Yield **** **(%)**	**Seed Yield *** **(%)**	
INIAP 2009	38.05 ± 2.25 c	43.10 ± 3.14 a	14.84 ± 0.88 a	
P10	36.59 ± 0.47 c	45.94 ± 0.89 a	17.57 ± 0.43 a	
Sweet passion fruit	84.34 ± 2.73 a	13.01 ± 2.54 b	2.53 ± 0.20 c	
Gulupa	47.10 ± 2.56 b	46.34 ± 2.34 a	6.56 ± 0.33 b	
Criollo POR1	34.54 ± 3.32 c	46.59 ± 5.73 a	18.87 ± 2.67 a	
Criollo PICH1	45.67 ± 1.71 b	46.75 ± 1.62 a	5.60 ± 0.32 bc	

Means with different letter are significantly different at 5% level. * fruit traits. ** pulp fruit traits.

**Table 2 plants-11-00697-t002:** Macro and micronutrient content in pulp of passion fruit germplasm grown in Ecuador. Values are expressed mg 100 g (DW) pulp^−1^.

Macronutrients							
**Genotype**	**N**	**K**	**P**	**Ca**	**Mg**	**S**	**Na**
INIAP 2009	818.68 ± 50.39 d	2456.05 ± 49.87 b	112.24 ± 0.15 e	26.93 ± 0.13 b	86.86 ± 5.30 c	81.62 ± 0.07 b	8.81 ± 0.07 e
P10	1218.99 ± 7.36 a	2816.00 ± 76.46 a	88.56 ± 0.64 f	18.82 ± 1.09 c	69.32 ± 3.95 d	73.52 ± 0.89 c	12.48 ± 0.11 a
Sweet passion fruit	925.93 ± 17.98 c	1471.28 ± 58.85 e	147.44 ± 0.30 b	8.00 ± 0.24 e	97.10 ± 0.48 c	139.71 ± 0.62 a	9.49 ± 0.06 cd
Gulupa	1101.05 ± 8.22 b	1926.84 ± 18.35 d	134.82 ± 0.25 c	19.70 ± 0.74 c	196.97 ± 7.34 a	41.96 ± 0.61 e	9.22 ± 0.07 de
Criollo POR1	1270.96 ± 13.11 a	2574.69 ± 6.56 b	131.12 ± 0.08 d	16.64 ± 0.13 d	100.76 ± 8.91 c	74.55 ± 1.18 c	10.05 ± 0.20 c
Criollo PICH1	1091.13 ± 12.69 b	2252.25 ± 19.04 c	170.03 ± 1.27 a	42.51 ± 0.13 a	160.85 ± 4.38 b	57.95 ± 1.97 d	11.00 ± 0.45 b
**Micronutrients**							
**Genotype**	**B**	**Zn**	**Cu**	**Fe**			
INIAP 2009	1.18 ± 0.07 b	2.12 ± 0.13 ab	0.13 ± 0.01 b	7.62 ± 0.15 b			
P10	1.21 ± 0.06 b	1.53 ± 0.13 bc	0.14 ± 0.01 b	4.19 ± 0.12 c			
Sweet passion fruit	0.53 ± 0.06 d	2.77 ± 0.18 a	0.24 ± 0.01 a	8.24 ± 0.59 b			
Gulupa	1.41 ± 0.06 a	2.69 ± 0.25 a	0.13 ± 0.01 b	4.40 ± 0.74 c			
Criollo POR1	1.11 ± 0.07 b	0.98 ± 0.68 c	0.12 ± 0.01 b	9.56 ± 0.40 a			
Criollo PICH1	0.82 ± 0.07 c	2.09 ± 0.07 ab	0.25 ± 0.01 a	8.08 ± 0.08 b			

Means with different letter are significantly different at 5% level.

**Table 3 plants-11-00697-t003:** Pearson correlation between fruit quality traits and mineral content of passion fruit germplasm. Bold represents statistical significance at 5%.

	Plant Yield	Fruit Weight	Peel Thickness	Pulp Yield	Soluble Solids	Titratable Acidity	Vitamin C	N	K	P	Ca	Mg	S	Na	B	Zn	Cu	Fe	Mn
**Plant yield**	1.00	0.68	0.49	0.32	−0.36	0.79	0.31	0.01	0.66	**−0.94**	−0.15	−0.57	−0.13	0.29	0.57	0.31	−0.65	−0.53	0.70
**Fruit weight**		1.00	**0.90**	−0.35	0.17	0.45	−0.40	−0.36	0.20	−0.67	−0.43	**−0.88**	0.60	0.05	−0.12	−0.11	−0.22	−0.01	0.47
**Peel Thickness**			1.00	−0.61	0.47	0.14	−0.60	−0.49	−0.13	−0.54	−0.60	**−0.75**	**0.76**	−0.08	−0.27	0.17	−0.06	−0.08	0.27
**Pulp yield**				1.00	**−0.88**	0.61	**0.92**	0.47	**0.75**	−0.24	0.57	0.26	**−0.90**	0.29	**0.74**	−0.51	−0.52	−0.27	0.33
**Soluble Solids**					1.00	−0.66	**−0.75**	−0.27	**−0.82**	0.20	**−0.79**	−0.04	**0.74**	−0.40	−0.51	0.50	0.26	0.12	−0.51
**Titratable Acidity**						1.00	0.53	0.19	**0.91**	**−0.75**	0.13	−0.55	−0.31	0.24	0.58	−0.69	−0.72	−0.07	0.64
**Vitamin C**							1.00	0.36	0.60	−0.24	0.45	0.39	**−0.94**	0.03	**0.87**	−0.34	−0.64	−0.32	0.09
**N**								1.00	0.46	−0.12	−0.04	0.06	−0.43	0.62	0.28	−0.62	−0.25	−0.13	0.33
**K**									1.00	−0.61	0.32	−0.41	−0.48	0.54	0.54	**−0.77**	−0.54	−0.13	**0.76**
**P**										1.00	0.37	0.61	0.05	−0.28	−0.56	0.35	0.72	0.51	−0.66
**Ca**											1.00	0.38	−0.59	0.19	0.09	−0.06	0.29	0.07	0.13
**Mg**												1.00	−0.59	−0.27	0.21	0.45	0.18	−0.20	−0.62
**S**													1.00	−0.17	−0.77	0.21	0.39	0.42	−0.08
**Na**														1.00	−0.01	−0.43	0.13	−0.33	**0.83**
**B**															1.00	−0.24	**−0.83**	−0.58	0.17
**Zn**																1.00	0.43	−0.24	−0.52
**Cu**																	1.00	0.28	−0.16
**Fe**																		1.00	−0.34
**Mn**																			1.00

**Table 4 plants-11-00697-t004:** Soil nutrient content (mg kg^−1^) of the passion fruit orchards in the INIAP´s research sites.

Site	N	P	K	Ca	Mg	S	Zn	Cu	Fe	Mn	B
Portoviejo	23 M	13 M	507 H	3400 H	864 H	23 H	3.5 M	6 H	56 H	25 H	0.6 M
Quevedo	23 M	20 M	273 H	2800 H	276 H	17 M	8.5 H	17 H	330 H	7.5 M	0.3 L
Tumbaco	42 M	86 H	273 H	1600 M	408 H	12 M	13 H	9 H	59 H	5.4 M	0.8 L

H, high; M, medium; L, low. These categories are based on nutrient ranges determined for Ecuadorian soils [80].

**Table 5 plants-11-00697-t005:** *Passiflora* species analyzed to determine the fruit quality traits and mineral content.

Species	Name	Type of Germplasm	Site	Province	Latitude (South)	Longitude (West)	Altitude (Masl)	Annual Precipitation (mm)	Annual Average Temperature (°C)	Heliophany (Hours/ Year)
*Passiflora edulis* f. *flavicarpa*	INIAP 2009	EV	Portoviejo	Manabí	01°09′43″	80°23′06″	52	852	26	1385
*Passiflora edulis* f. *flavicarpa*	P10	BG								
*Passiflora* sp.	Criollo POR1	ELG								
*Passiflora* sp.	Sweet PF	IG	Quevedo	Los Ríos	01°04′24″	79°29′14″	74	1200	25	920
*Passiflora edulis* f. *edulis*	Criollo PICH1	ELG								
*Passiflora alata*	Gulupa	IG	Tumbaco	Pichincha	00°12′57″	78°24′43″	2348	892	17	2039

EV, Ecuadorian variety; BG, breeding germplasm; ELG, Ecuadorian local germplasm; IG, introduced germplasm.

## Data Availability

Data are contained within the article.

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
