# Peer review of "Pulp Mineral Content of Passion Fruit Germplasm Grown in Ecuador and Its Relationship with Fruit Quality Traits"

_plants, 2022, doi:10.3390/plants11050697_

Round 1

Reviewer 1 Report

The manuscript describes the leaf mineral content in pulp of different passion fruit genotypes and species, when growing under pedo-climatic conditions of Ecuador. The manuscript reports pulp mineral nutrition for seven macronutrients (N, P, K, Ca, Mg, S and Na) and five micronutrients (B, Zn, Cu, Fe and Mn), as well as fruit weight, peel thickness, soluble solids content, acidity and vitamin C, during one season.

However, it does not report, plant vigour or development and fruit yield parameters that could contribute to have a best idea of the agronomical performance of the studied plant materials in the climate and soil conditions considered.

My main concern is the limited number of seasons (one season) considered for mineral nutrients and fruit quality and absence of other agronomic parameters related with plant vigour and yield efficiency. Similarly, authors should have included very important agronomic parameters as blooming and harvesting times, plant development and yield. They could help to explain cultivars performance, especially if only a season is considered for the evaluated traits.

In addition, it is strongly advised to update bibliography and review similar works with other fruit studies. Authors will see how different factors as yield, vigour and years of evaluation are affecting mineral nutrition and other physiological traits. It will add scientific relevance and increase the interest of results for a wider number of readers and farmers.

Some additional remarks:

In MATERIALS AND METHODS, number and characteristics/definition of assessed plants per each genotype should be more precisely described.

For instance: How many plants and/or replicates have been tested per genotype? Are all of them growing in the same place? Do they have replicates? Which is the experimental design followed to take samples and select the assessed plants? Other agronomic parameters as plant habit and vigour, and yield are strongly recommended to be included.

In the Experimental section, authors should also explain how the evaluated genotypes were obtained. Were they propagated sexually by seeds or clonally by cuttings? This information must be included in Table 4. In fact, it is not clear if the tested materials are released cultivars, unselected local varieties, or accessions maintained in a germplasm collection. Please, see Lines 156-157 and clarify: released cultivars, varieties, genotypes, accessions?

It is also strongly advised to improve the RESULTS and DISCUSSION section because it is very repetitive for each trait including numerical values both in the text and also in the respective Tables.

Could authors say if the lower values for several traits could be due to plant vigour and total yield?.  Lack or excess of plant vigour and yield could be other factors to explain differences and it should be discussed.

Authors should also review English grammar because there are numerous mistakes (*) and sentences too extended (**). It is difficult to read and understand those sentences.

(*) For instance, see Line 336… this result is not agree with… Line 340, … which is agree with…; Line 344, …which are agree…

(**) For instance, see the first paragraph in the Introduction section (lines 31-40), sentences between lines 359 and 364, as well as between lines 85 and 88. Meaning is no clear.

Authors must follow the Journal rules to include REFERENCES in the text, because they seem to be duplicated when mentioned.

Author Response

Reviewer 1

Comment: it does not report, plant vigour or development and fruit yield parameters that could contribute to have a best idea of the agronomical performance of the studied plant materials in the climate and soil conditions considered. Other agronomic parameters as plant habit and vigour, and yield are strongly recommended to be included. Could authors say if the lower values for several traits could be due to plant vigour and total yield?.  Lack or excess of plant vigour and yield could be other factors to explain differences and it should be discussed. In addition, it is strongly advised to update bibliography and review similar works with other fruit studies. Authors will see how different factors as yield, vigour and years of evaluation are affecting mineral nutrition and other physiological traits. It will add scientific relevance and increase the interest of results for a wider number of readers and farmers.

Reply: Average yield per plant has been added but data of vigor was not recorded in this study because this study was focus in the mineral composition of the fruit pulp and its relationship with fruit quality traits that is why variables such as fruit weight was mentioned in the original manuscript. Plant did not show any symptom of nutrient deficiency in the field and fruit weight and yield per plant are similar to those reported in other studies thus plant growth was adequate, this has been mentioned in the discussion. Only Gulupa showed less yield than that reported in Colombia but this trait can vary due to environmental conditions. The correlation of plant yield and the minerals was also estimated and the main results are discussed. It has been mentioned that further studies taking into consideration foliar analysis to know the mineral content in this plant organ should be made to have a better understanding about lant mineral content and their relationship with variables such as yield. Literature for other fruit crops is included in the discussion. In fact, we agree that yield, vigour and years of evaluation can influence mineral content but this relationship can be better understood with foliar mineral analysis than pulp analysis, for this reason we added this point like a recommendation.

Comment: My main concern is the limited number of seasons (one season) considered for mineral nutrients and fruit quality and absence of other agronomic parameters related with plant vigour and yield efficiency. Similarly, authors should have included very important agronomic parameters as blooming and harvesting times, plant development and yield. They could help to explain cultivars performance, especially if only a season is considered for the evaluated traits.

Reply: It has been mentioned that the plants were transplanted in August 2020 y harvesting in February 2021 (this time is a transition between the dry and rainy season). Due the geographical position of Ecuador, the country do not have the four seasons like other countries in different latitudes. In Ecuador there is some variation in the rainy and dry season but the main factor for the growing of passion fruit is the temperature which has slight variations during the year and the average temperature is almost constant. For this reason, this fruit crop can be sown and harvesting any time of the year due to that the temperature is not a limitation. However, in this fruit species there is an interaction environment x genotype that has been mentioned in the discussion section.

Comment: In MATERIALS AND METHODS, number and characteristics/definition of assessed plants per each genotype should be more precisely described. For instance: How many plants and/or replicates have been tested per genotype? Are all of them growing in the same place? Do they have replicates? Which is the experimental design followed to take samples and select the assessed plants?

Reply: This information has been added in the section 3.1.

In the Experimental section, authors should also explain how the evaluated genotypes were obtained. Were they propagated sexually by seeds or clonally by cuttings? This information must be included in Table 4. In fact, it is not clear if the tested materials are released cultivars, unselected local varieties, or accessions maintained in a germplasm collection. Please, see Lines 156-157 and clarify: released cultivars, varieties, genotypes, accessions?

Reply: Table 4 indicates the location where the genotypes were obtained. It was clarified that the assessed plants were propagated by cuttings in section 3.1. It was explained more in detail the genotypes to clarify them.

It is also strongly advised to improve the RESULTS and DISCUSSION section because it is very repetitive for each trait including numerical values both in the text and also in the respective Tables.

Reply: the values in the text has been deleted to avoid repeatability and just the comparison to the results with others studies is mentioned in the discussion.

Comment: Authors should also review English grammar because there are numerous mistakes (*) and sentences too extended (**). It is difficult to read and understand those sentences. (*) For instance, see Line 336… this result is not agree with… Line 340, … which is agree with…; Line 344, …which are agree… (**) For instance, see the first paragraph in the Introduction section (lines 31-40), sentences between lines 359 and 364, as well as between lines 85 and 88. Meaning is no clear.

Reply: The sentences corrected and paragraphs were rewritten to make their meaning clear. The English grammar was checked by Dr. Tissa Kannagara (from Canada) as is mentioned in the acknowledgment.

Comment: Authors must follow the Journal rules to include REFERENCES in the text, because they seem to be duplicated when mentioned.

Reply: This has been corrected in the whole document.

Reviewer 2 Report

In the present manuscript, the mineral content in the pulp of different germplasm of passion fruit (Passiflora edulis f. flavicarpa, P. alata, P. edulis f. edulis and Passiflora sp.) grown in Ecuador was evauated.

More details about the sampling of fruit are neede.

How the randomized complete design was performed in plnat located at different regions?

The discussion should be deeper and try to reveal the reasons why the observed correlations were recorded.

Author Response

Reviewer 2

Comment: More details about the sampling of fruit are needed.

Reply: It was clarified that sampling was carried out in the orchards belonging to the research sites of INIAP. Each replicate was constituted by 25 fruits in total (five from five different plants from the same orchard) harvested at maturity, and then pulp of all fruits were obtained, homogenized and lyophilized for the analysis. Fruits were evaluated directly for the physical traits and 10 g of the lyophilized sample was used for the mineral analysis.

Comment: How the randomized complete design was performed in plant located at different regions?

Reply: This point was corrected and clarified in the statistical analysis section. It was mentioned that each different germplasm was set as a treatment without considering the place of origin because the objective was to compare among germplasm to know what were the differences in terms of physical traits and mineral content among them.

Comment: The discussion should be deeper and try to reveal the reasons why the observed correlations were recorded.

Reply: The discussion about the correlation results has been improved, more literature has been included to explain the correlation results.

Reviewer 3 Report

The manuscript plants-1592770 addresses the quality traits and Mineral Content of purple, yellow and sweet Passion Fruit Germplasm Grown in Ecuador. The topic is interesting for growers, consumers and for breeding purposes or research. The manuscript covers the main background literature about the topic, the authors extensively described the main quality traits of each germplasm, concluding that there are some nutrients, particularly Mg, that could be further explored in fertilization programs due to its negative association to peel thickness. Overall the paper is well written, but there are a few issues that should be clarified to publish the article in Plants. Recommendation: accept with minor revisions. Please see the attached file for details.

Author Response

Reviewer 3

Comment: Please see the attached file for details.

Reply: All the correction suggested in the attached fila have been made in the new version of the manuscript. Paragraphs were rephrasing, some words changed according the reviewer suggestion and some words deleted.

Comment: add "purple, yellow, and sweet" in the title.

Reply: This words were added in the title.

Comment: How is P content associated with fruit delayed senescence and how this information could be related to the results obtained by the authors? Please explain and rephrase.

Reply: There was a mistake in the information because P is related to leaf senescence, for this reason the sentence was deleted.

Comment: Please add a reference study regarding the sensory properties of Gulupa.

Reply:  The reference of Bermeo (2020) “Evaluation of the influence of the degree of maturity of gulupa (Passiflora edulis Sims) on sensory acceptance in food products” was added in the paragraph.

Comment: Please try to report other studies in the same species; otherwise I suggest to clarify the plant material used in other studies.

Reply: The species were clarified in all cases.

Comment: There are several aspects that should be included: when was the experiment conducted (year)?How many fruits were sampled per germplasm? what was the harvest date?

Reply: It was indicated that the experiment was conducted in 2021, the harvest date was in November 2020 and that 25 fruits were harvested (five fruits from different five plants in the orchard).

Comment: This part should move to sub-section 3.1

Reply: The information was moved to section 3.1 Experimental site and plant material

Reviewer 4 Report

The manuscript entitled "Quality Traits and Mineral Content of Passion Fruit Germplasm Grown in Ecuador" by Viera et al. provides an insight into the quality characteristics of passion fruit germplasm from Ecuador.
The manuscript is very well written and easy to follow.
In general, it is a screening work, with few varieties, that focuses mainly upon the concentration of elements within the fruit and basic quality parameters. It would be advised to include a greater number of known native cultivars that exist in Ecuador, and also to include measurements that focus upon the antioxidant activity of the pulp or juice, as well as the phenolic total content and profile of the genetic material. The antioxidant value of these fruits is also very important for the consumers and there is a necessity to search and elucidate. Try to perform at least two antioxidant methods to estimate the total phenolic content, and provide the phenolic profile of each accession.
Overall, the manuscript needs further measurements in order to be worthy of being published in Plants.

Author Response

Reviewer 4

Comment: It would be advised to include a greater number of known native cultivars that exist in Ecuador, and also to include measurements that focus upon the antioxidant activity of the pulp or juice, as well as the phenolic total content and profile of the genetic material. The antioxidant value of these fruits is also very important for the consumers and there is a necessity to search and elucidate. Try to perform at least two antioxidant methods to estimate the total phenolic content, and provide the phenolic profile of each accession. 

Reply: The genotypes used for this study correspond to the germplasm which is the ex situ germplasm bank of INIAP and they will be the vegetal material that will be used for breeding works in this fruit crop, such as further interspecific crosses and progeny evaluation. More details about the germplasms have been added in the section 3.1.

The antioxidant characterization of the passion fruit germplasm used in this study was already carried out and is published in the following reference “Phytochemical composition and antioxidant activity of Passiflora spp. germplasm grown in Ecuador. Plants 2022, 11, 328.”; consequently, this information complement to the antioxidant characterization. The main results of antioxidant characterization have been mentioned in the discussion (end of the section 2.1) to stand out the results of the germplasms in addition to the mineral content. How the reference has been cited if the reader wants to obtain more detailed information will be able to consult the published article.

Round 2

Reviewer 1 Report

The manuscript describes the mineral content in pulp of 6 passion fruit genotypes, when growing under pedo-climatic conditions of Ecuador. It reports pulp mineral nutrition for seven macronutrients (N, P, K, Ca, Mg, S and Na) and five micronutrients (B, Zn, Cu, Fe and Mn), as well as yield, fruit weight, peel thickness, soluble solids content, acidity and vitamin C, during one season (2021).

My main concern is the limited number of seasons (one season, year 2021) considered for fruit quality parameters and even more, the experimental design followed to take samples and perform their analysis.

For instance (L 431-433), it is unclear if biological or technological replicates were finally analysed. In fact, it seems to me that authors have only analysed 3 technical replicates. They say that they harvested 5 fruits from 5 individual plants to obtain a total of 25 fruits. However, it seems that all fruits were finally mixed to obtain 10 grams of freeze-dried pulp and after that, only 3 technical replicates (n=3) were selected for mineral nutrients analysis. In this case, this study is not valid and should be performed during additional harvesting seasons. This type of analysis must be done with biological replicates, corresponding to individual plants adequately labelled during all the procedures, especially if authors report data only from a single harvesting season.

Authors should also review English grammar because there are numerous mistakes. For instance see Line 181: the latter values is lower that the reported by …

In addition, it is difficult to read and understand extended sentences in some cases. For instance, meaning is not clear in sentences between Lines 104-108 and Lines 109-117.

Some additional remarks:

Authors must follow the Journal rules to use references in the text. It is strongly advised to see Journal Guidelines and other articles published in the Journal. For instance see L53: In addition, (2) reported that …. It should be written: In addition, Santos et al (2) reported … Similar remark in Lines: L 113, by (28) in Colombia, and L 118, to that reported by (29) with …

Authors should use more appropriate terms for the genotypes/accessions/varieties and cultivars used in their work. It is not clear. For instance, in Line 56, ‘Criollo’ seems to be a local variety where variability and different characteristics can be potentially found. A cultivar should have a well characterised genotype, always showing same features and clearly identified by farmers. 

Lines 246-260. Authors report numerical values for polyphenols and antioxidant activity previously published. It is very repetitive and it is not necessary, because values have been already published.

Table 5: According to the Journal rules and International system of Units, review if ppm and meq/100mL are used as appropriated Units. In addition, it is not clear which unit correspond to which mineral element. Letters H, M and H should also describe specific values according to a precise scale or numerical range.

Author Response

Reviewer 1

Comment: My main concern is the limited number of seasons (one season, year 2021) considered for fruit quality parameters and even more, the experimental design followed to take samples and perform their analysis.

Reply: In fact, the study was carried out just in one season considering that the temperature which is the most important factor for the development of passion fruit remains without much variation in Ecuador because we do not have the four seasons (in terms of environment conditions). We understand your concern and it is mentioned that this study is a reference for further research, including more aspects such genotype x environment interaction. Each genotype was considered as a treatment and the means of the distinct genotypes were compared to find differences between them. 

Comment: For instance (L 431-433), it is unclear if biological or technological replicates were finally analyzed. In fact, it seems to me that authors have only analyzed 3 technical replicates. They say that they harvested 5 fruits from 5 individual plants to obtain a total of 25 fruits. However, it seems that all fruits were finally mixed to obtain 10 grams of freeze-dried pulp and after that, only 3 technical replicates (n=3) were selected for mineral nutrients analysis. In this case, this study is not valid and should be performed during additional harvesting seasons. This type of analysis must be done with biological replicates, corresponding to individual plants adequately labelled during all the procedures, especially if authors report data only from a single harvesting season.

Reply: It has been clarified that for the mineral analysis the process of collecting 5 fruits for 5 individuals was done by triplicate using different plants because the replicates (n=3) were independent samples (biological replicates).

Comment: Authors should also review English grammar because there are numerous mistakes. For instance see Line 181: the latter values is lower that the reported by …

Reply: The English grammar was checked by Dr. Tissa Kannagara (from Canada) and it is mentioned in the acknowledgment.

Comment: In addition, it is difficult to read and understand extended sentences in some cases. For instance, meaning is not clear in sentences between Lines 104-108 and Lines 109-117.

Reply: These paragraphs have been re-written.

Comment: Authors must follow the Journal rules to use references in the text. It is strongly advised to see Journal Guidelines and other articles published in the Journal. For instance see L53: In addition, (2) reported that …. It should be written: In addition, Santos et al (2) reported … Similar remark in Lines: L 113, by (28) in Colombia, and L 118, to that reported by (29) with …

Reply: Text citations has been corrected based on the journal´s guideline and other published articles.

Comment: Authors should use more appropriate terms for the genotypes/accessions/varieties and cultivars used in their work. It is not clear. For instance, in Line 56, ‘Criollo’ seems to be a local variety where variability and different characteristics can be potentially found. A cultivar should have a well characterized genotype, always showing same features and clearly identified by farmers. 

Reply: The only released variety of passion fruit in Ecuador is INIAP 2009 which has been characterized prior its release. The local passion fruit germplasms are grown but they have not been characterized consequently they would be genotypes. This has been corrected in the manuscript.

Comment: Lines 246-260. Authors report numerical values for polyphenols and antioxidant activity previously published. It is very repetitive and it is not necessary, because values have been already published.

Reply: The information about polyphenols and antioxidant activity was included because another reviewer asked it. However, according to the suggestion, numerical values have been deleted.

Comment: Table 5: According to the Journal rules and International system of Units, review if ppm and meq/100mL are used as appropriated Units. In addition, it is not clear which unit correspond to which mineral element. Letters H, M and H should also describe specific values according to a precise scale or numerical range.

Reply: The values for the nutrients indicated in table 5 have been expressed in mg kg-1 to uniform the units. It was clarified that the categories (H, M and L) are based on the ranges determined for Ecuadorian soils, the reference that is used by the Soil laboratory to do the interpretation has been added.

Reviewer 2 Report

The authors addressed most of my comments.

However, considering that the evaluated germplasm was cultivated at different sites does not allow to make safe conclusions regarding the effect of genotype on the studied parameters.

Author Response

Comment: However, considering that the evaluated germplasm was cultivated at different sites does not allow to make safe conclusions regarding the effect of genotype on the studied parameters.

Reply: This limitation has been mentioned in the last paragraph of the Result and Discussion section, mentioning that the passion fruit germplasms were grown in different Ecuadorian environmental conditions (altitude, precipitation and heliophany) because of their preference in adaptation for their growth and production, and it is also mentioned that these results are a reference for further studies considering the genotype x environment interaction.

Reviewer 4 Report

The manuscript is worthy of being published in its present form

Author Response

Comment: The manuscript is worthy of being published in its present form.

Reply: the authors thanks to the reviewer for his/her comments to improve the manuscript.